# Scalable Gas Sensing, Mapping, and Path Planning via Decentralized Hilbert Maps

**DOI:** 10.3390/s19071524

**Published:** 2019-03-28

**Authors:** Pingping Zhu, Silvia Ferrari, Julian Morelli, Richard Linares, Bryce Doerr

**Affiliations:** 1Sibley School of Mechanical and Aerospace Engineering, Cornell University, Ithaca, NY 14853, USA; ferrari@cornell.edu (S.F.); jam888@cornell.edu (J.M.); 2Department of Aeronautics and Astronautics, Massachusetts Institute of Technology, Cambridge, MA 02139, USA; linaresr@mit.edu; 3Department of Aerospace Engineering and Mechanics, University of Minnesota, Minneapolis, MN 55455, USA; doerr024@umn.edu

**Keywords:** gas sensing, multi-agent systems, very large-scale robotic systems, information theory

## Abstract

This paper develops a decentralized approach to gas distribution mapping (GDM) and information-driven path planning for large-scale distributed sensing systems. Gas mapping is performed using a probabilistic representation known as a Hilbert map, which formulates the mapping problem as a multi-class classification task and uses kernel logistic regression to train a discriminative classifier online. A novel Hilbert map information fusion method is presented for rapidly merging the information from individual robot maps using limited data communication. A communication strategy that implements data fusion among many robots is also presented for the decentralized computation of GDMs. New entropy-based information-driven path-planning methods are developed and compared to existing approaches, such as particle swarm optimization (PSO) and random walks (RW). Numerical experiments conducted in simulated indoor and outdoor environments show that the information-driven approaches proposed in this paper far outperform other approaches, and avoid mutual collisions in real time.

## 1. Introduction

Fugitive emissions and the dispersion of pollutants in the atmosphere are significant concerns affecting public health as well as climate change. The accidental release of hazardous gases from both urban and industrial sources is responsible for a variety of respiratory illnesses and environmental concerns [1,2]. Many sensors are currently fabricated and deployed both indoors and outdoors for air quality data collection and communication. However, obtaining a spatial representation of a gas distribution is a challenging problem because existing mapping and fusion algorithms do not scale to networks of potentially hundreds of mobile and stationary sensors. The decentralized mapping and path-planning methods presented in this paper are applicable to very large-scale sensor systems, and as such can potentially be used to assess air quality, classify safe and hazardous areas based on the concentration of the harmful gases, and even localize fugitive emissions [3].

Due to the fundamental mechanisms of atmospheric gas dispersion, auxiliary tools are required to ensure early detection and to respond with appropriate counter measures by planning future measurements efficiently in both space and time. Methods for obtaining gas distribution maps (GDM) have recently been developed along two lines of research [4]. In the first line of research, a stationary sensor network is used to collect and fuse measurements to estimate the position of a source, typically requiring expensive and time-consuming calibration and data-recovery operations [5]. The use of stationary sensors, however, is not typically effective or sufficiently expedient for gas sensing in response to high source rates of critical emissions. The second line of research, pursued in this paper, involves the use of mobile sensors, such as terrestrial robots equipped with gas sensors that can be controlled to rapidly and efficiently collect and fuse measurements over time. The latter approach provides for flexible sensor configurations that can respond to measurements online and thus can be applied to mapping gas distributions in unknown and complex environments.

Mobile gas sensing can be performed by a single robot [2,3,6,7,8,9] or by networks of robots also referred to as multi-agent systems (MAS) [10,11,12,13,14]. Compared to single robots, MAS present several advantages, including increased probability of success and improved overall operational efficiency [14], but also present additional technical challenges. In addition to requiring solving the GDM problem as a dynamical optimization problem, multi-agent path planning, coordination, communication, and fusion can become intractable as the number of robots increases [4,15,16,17,18,19,20].

MAS path planning and coordination can be achieved via centralized or decentralized methods, depending on the underlying communication infrastructure [21]. Centralized methods require persistent communication between a central station and every agent in the network, such that the central station can process and fuse all sensor measurements and use them to plan, or re-plan, the robot paths in a coordinated and collaborative fashion. Decentralized methods allow each robot to process its own measurements individually and then to communicate and fuse it with the measurements of a subset of collaborative robots, such as its nearest neighbors, with established connectivity. As a result, the performance of centralized methods depends entirely on the reliability of communication protocols under the operating conditions. Because fragile communication links are common in many hazardous scenarios, reliance on persistent communications with the central station and the associated power consumption can hinder or even prevent the applicability of MAS over large regions of interest that require repeated long-distance data transmission. Nevertheless, most of the existing GDM methods to date rely on centralized methods and algorithms [10,12,13].

One of the main challenges in developing decentralized GDM methods lies in solving the data fusion problem for neighboring robots such that each robot can build its own representation of the GDM based on local measurements, but also update it incrementally as new information is obtained from neighboring robots. Considering the limited communication bandwidth and computing resources of most autonomous robots, it is also impractical to expect each sensor to share all the raw measurement data with its neighbors. Therefore, the decentralized approach to gas source localization presented in [11] shares only the largest gas concentration and corresponding source position with its neighbors. However, this decentralized approach cannot be extended to high-performance GDM representations, such as Gaussian process (GP) mixture models [8] or kernel-based models [3,22], and fusing GDMs obtained by different robots would result in redundant and expensive computations that potentially operate on the same raw data repeatedly.

The Hilbert map is an alternate GDM that represents a probability map learned from local concentration measurements [7,23]. The novel GDM representation developed in this paper uses kernel logistic regression (KLR) to express the probability that the gas concentration at a certain position belongs to a predefined range as a Hilbert map function. By this approach, new decentralized fusion algorithms can be developed that present several advantages, including decentralized fusion, agent-level complete GDM representation, and update for decentralized decision making. Additionally, information fusion operations are implemented via simple summations and only the local Hilbert map needs to be shared among neighboring robots at every measurement update. As a result, at the end time of the task, information about the gas concentration distribution over the entire region of interest can be delivered to the client or operator even if only a few robots complete the mission. Two GDM-based path-planning methods, the entropy-based artificial potential field (EAPF) and the entropy-based particle swarm optimization (EPSO) algorithms, are presented in this paper, and the simulation results show that they significantly outperform existing algorithms.

## 2. Problem Formulation

Consider the problem of optimally planning the trajectory of a distributed sensing system comprised of *N* cooperative robots engaged in GDM through a large region of interest (ROI), denoted by $W\subset R^{2}$. In general, the gas concentration at a position x∈W, at time *t*, is modeled as a random variable C(x,t), thus the whole gas concentration is a random field. The gas distribution is considered constant over a time interval for simplicity and thus is modeled as a spatial function c(x) defined over W. The approach can be extended to time-varying concentrations by augmenting the dimensionality of the GDM.

Let $U\subset R^{m}$ denote the space of admissible actions or controls. The dynamics of each robot are governed by stochastic differential equations (SDEs),
(1)$${\dot{x}}_{n}\left(t\right)=f\left[x_{n}\left(t\right),u_{n}\left(t\right),t\right]+\mathrm{Gw}\left(t\right),\;x_{n}\left(T_{0}\right)=x_{n_{0}}\;\mathrm{and}\;n=1,\ldots ,N$$
where $x_{n}\left(t\right)\in W$ denotes the *n*th robot’s position and the velocity at time *t*, $u_{n}\left(t\right)\in U$ denotes the *n*th robot action or control, and $x_{n_{0}}$ denotes the robot initial conditions at initial time $T_{0}$. The robot dynamics in (1) are characterized by additive Gaussian white noise, denoted by $w\left(t\right)\in R^{2}$, and $G\in R^{2\times 2}$ is a known constant matrix. In this paper, we assume that the position of the *n*th robot is obtained by an on-board GPS and is denoted by ${\hat{x}}_{n}\left(t\right)$, where $\left(\hat{\cdot }\right)$ denotes the estimated variable’s value.

All *N* robots are equipped with identical metal oxide sensors in order to cooperatively map a time-constant gas distribution, where the quantity and position of gas plumes are unknown *a priori*. Because the reaction surface of a single gas sensor is very small (≈1 cm${}^{2}$), a single measurement from a gas sensor can only provide information about a very small area. Therefore, to increase the resolution of the GDM, a small metal oxide sensor array is used instead of a single oxide sensor for each robot. The area that is covered by this small metal oxide sensor array is called the field of view (FOV) of the robot, denoted by $S(x_{n})\in W$. Because the structure of the metal oxide sensor array is fixed, the positions of each metal oxide sensor can be approximated by the robot position measurements, denoted by ${\hat{x}}_{n,m}$, m=1:M, where *M* is the number of oxide sensors on-board the *n*th robot. For example, for a 5×5 metal oxide sensor array, the number of sensors is M=25.

The gas concentration measurement obtained by the *m*th sensor can be modeled as the sum of the actual concentration and measurement noise,
(2)$${\hat{c}}_{n,m}\left(x_{n}\right)=c\left(x_{n,m}\right)+\hat{v}\left(x_{n,m}\right)$$
where $\hat{v}\left(x_{n,m}\right)$ is a realization of the random measurement noise, $V(x_{n,m})$. Furthermore, the concentration measurement, ${\hat{c}}_{n,m}\left(x_{n}\right)$, can be treated as a sample of the random variable $C(x_{n,m})$, defined as

(3)$$C\left(x_{n,m}\right)≜c\left(x_{n,m}\right)+V\left(x_{n,m}\right)$$

Then, the concentration measurement ${\hat{c}}_{n,m}\left(x_{n}\right)$ can be considered as an observation of the random field C(·) at position $x_{n,m}$.

Because the actual gas concentration, c(x), and the distribution of the measurement noise, V(x), are unknown, our objective is to approximate the random field C(·), and then use this approximated random field to estimate the gas concentration map. The cost function of the *n*th robot over a fixed time interval $\left[t_{0},\;t_{f}\right]$ can then be expressed as an integral of the future measurement values, conditioned on past measurements, and the robot control usage, i.e.,
(4)$$J_{n}=\int_{W}H\left[C_{n}\left(x|M_{n}\left(T_{f}\right)\right)\right]dx+\int_{t_{0}}^{t_{f}}u_{n}{\left(t\right)}^{T}Ru_{n}\left(t\right)dt$$
where $H[C_{n}\left(x|M_{n}\left(t\right)\right)]$ is the entropy of the random variable $C_{n}\left(x|M_{n}\left(t\right)\right)$ approximated by the *n*th robot given all past measurements, $M_{n}$. R is a positive definite matrix that weighs the importance of the elements of the control input $u_{n}\left(t\right)$, and the superscript “$T^{}$” denotes the matrix transpose. The optimal planning problem is to find the optimal time history of the robot state $x_{n}\left(t\right)$ and control $u_{n}\left(t\right)$, for all n=1,…,N, so that the cost function $J_{n}$ in (4) is minimized over $\left[t_{0},t_{f}\right]$, and subject to (1).

Let the time interval $\left[t_{0},\;t_{f}\right]$ be discretized into $T_{f}$ equal time steps $\Delta t=(t_{f}-t_{0})/T$ and let $t_{k}=t_{0}+k\Delta t$ denote the discrete-time index with $k=1,\ldots ,T_{f}$. Then the objective function, $J_{n}$, can be rewritten as,
(5)$$J_{n}=\int_{W}H\left[C_{n}\left(x|M_{n}^{\left(T_{f}\right)}\right)\right]dx+\sum_{k=1}^{T_{f}}u_{n,k}^{T}Ru_{n,k}$$
where the subscript *k* indicates the *k*th time step and the superscript “${\left(T_{f}\right)}^{}$” indicates the time until the $T_{f}$th step. The term, $M_{n}^{\left(T_{f}\right)}=\overset{T_{f}}{\underset{k=1}{\cup }}M_{n,k}$, indicates the measurement history until the $T_{f}$th step and $M_{n,k}$ denotes all the measurements obtained by the *n*th robot up to the *k*th time step.

## 3. Representation of GDM

To approximate the probability distribution of the continuous random variable $C_{n}\left(x|M_{n}^{\left(T\right)}\right)$, there are several methods, including kernel density estimation (KDE) [24,25] and Gaussian process regression (GPR) [26]. In the KDE method, the learned probability density function (PDF) is assumed to be a weighted summation of many parameterized kernel functions, where the weight coefficients and the kernel parameters are critical and learned from the raw measurement data set. In the GPR method, the approximated PDF is assumed to be a Gaussian distribution, where a large matrix, the Gram matrix, is very critical and learned from the raw measurement data. For both of methods, the data fusion of two different measurement data sets and update of the approximated PDF are computationally demanding, because all computations must be implemented from a massive amount of raw data. It is difficult to obtain the updated parameters and coefficients from the previous parameters and coefficients directly.

The method in this paper overcomes this hurdle through an efficient fusion approach developed by approximating the continuous probability distribution by a discrete probability distribution. Denote the range of the concentration in the whole ROI by $R=[L_{c},H_{c}]$, and then divide the range into *L* concentration intervals denoted by $R_{1}=\left[L_{1},H_{1}\right),\ldots ,R_{l}=\left[L_{l},H_{l}\right),\ldots ,R_{L}=\left[L_{L},H_{L}\right]$, where $H_{l}=L_{l+1}$ for l=1,…,L-1, and $L_{1}=L_{c}$ and $H_{L}=H_{c}$. The cutoff coefficients, ${\left\{\left(L_{l},H_{l}\right)\right\}}_{l=1}^{L}$, specify the concentration intervals of interest. Instead of approximating the PDF, this paper models the probabilities that the gas concentration at every position x∈W belongs to the concentration interval ${\left\{p_{l}\left(x\right)\right\}}_{l=1}^{L}$, or,
(6)$$p_{l}\left(x\right)≜P\left(C\left(x\right)\in R_{l}\right),\;l=1,\ldots ,L$$
where P(·) denotes the probability operator. Let $f_{c}\left(x\right)$ denote the PDF of the concentration C(x) and $\Delta L_{l}=H_{l}-L_{l}$ denote the length of the *l*th concentration interval. The relationship between $p_{l}\left(x\right)$ and $f_{c}\left(x\right)$ can be expressed as,
(7)$$p_{l}\left(x\right)≜P\left(C\left(x\right)\in R_{l}\right)=\int_{x\in R_{l}}f_{c}\left(x\right)dx$$
where
(8)$$f_{c}\left(L_{l}\right)=\lim_{\Delta L\to 0}\frac{p_{l}\left(x\right)}{\Delta L}$$

Therefore, the discrete distribution ${\left\{p_{l}\left(x\right)\right\}}_{l=1}^{L}$ can be used to describe the probability distribution of the continuous random variable C(x) as L→∞. More importantly, it can be used for decision making, where the concentration intervals correspond to different levels of hazardous gas concentration. The Hilbert map method, developed by Fabio Ramos and Lionel Ott in [27] to model obstacle occupancy, is modified here to approximate the discrete distribution ${\left\{p_{l}\left(x\right)\right\}}_{l=1}^{L}$ over the entire ROI.

### 3.1. Mapping with KLR

A Hilbert map is a continuous probability map developed by formulating the mapping problem as a binary classification task. Let x∈W be any point in W and Y∈{0,1} be defined as a categorical random variable such that
(9)$$Y=\left\{\begin{array}{cc} 1, & \mathrm{if}\;C(x)\in R\\ 0, & \mathrm{if}\;C(x)\notin R \end{array}\right.$$
where the realization is denoted by *y*. The Hilbert map describes the probabilities P(Y=1|x)=p(x) and P(Y=0|x)=1-p(x) at the position x.

Consider the training measurement data set, $M={\left\{\left(x_{m},y_{m}\right)\right\}}_{m=1}^{M}$, where $x_{m}\in W$ indicates the measurement position, the Boolean variable $y_{m}\in \left\{1,0\right\}$ indicates whether the concentration measurement, $\hat{c}\left(x_{m}\right)$, belongs to the range, R (where 1=Yes and 0=No), and *M* is the number of measurements. The probability P(Y|x) is defined as,
(10)$$p\left(x\right)=P\left(Y=1|x\right)=1-\frac{1}{1+\exp\left[f(x)\right]}=\frac{e^{f(x)}}{1+e^{f(x)}}$$
where,
(11)$$1-p\left(x\right)=P\left(Y=0|x\right)=\frac{1}{1+e^{f(x)}}$$
and f∈H is a Hilbert function defined on W. H is a reproducing kernel Hilbert space (RKHS) associated with a kernel k(·,·). The kernel mapping is denoted by k(x,·)=φ(x),φ(x)∈H, and is an injective map from W to H [28,29,30,31,32]. According to the kernel trick [28,29,30,31,32], the evaluation of the Hilbert function can be expressed in the form of inner product,
(12)$$f\left(x\right)={\langle f,\varphi (x)\rangle }_{H}$$
where ${\langle \cdot ,\cdot \rangle }_{H}$ indicates the inner product in the RKHS. To learn the Hilbert map, the loss function $J_{H}$ is defined as,
(13)$$\begin{array}{ccc} J_{H} & = & \sum_{m=1}^{M}\ell_{m}+\frac{\lambda }{2}{∥f∥}_{H}^{2}\\ & = & \sum_{m=1}^{M}\left\{\ln\left[1+e^{f(x_{m})}\right]-y_{m}f\left(x_{m}\right)\right\}+\frac{\lambda }{2}{∥f∥}_{H}^{2} \end{array}$$
where $\ell_{m}=-\ln\left[P(Y=y_{m}|x_{m})\right]$ is the negative log-likelihood (NLL) of the data $\left(x_{m},y_{m}\right)$. Here, the term λ is the regularization term, which is a small user-defined positive scalar, λ≪1. Then the gradient of the loss function with respective to *f* is expressed as,
(14)$$\begin{array}{ccc} g & = & \frac{\partial J_{H}}{\partial f}=\sum_{m=1}^{M}\left[\frac{e^{f(x_{m})}}{1+e^{f(x_{m})}}\varphi \left(x_{m}\right)-y_{m}\varphi \left(x_{m}\right)\right]+\lambda f\\ & = & \sum_{m=1}^{M}\left[P\left(Y=1|x_{m}\right)-y_{m}\right]\varphi \left(x_{m}\right)+\lambda f\\ & = & \Phi (p-y)+\lambda f \end{array}$$
where $\Phi =\left[\varphi \left(x_{1}\right),\ldots ,\varphi \left(x_{M}\right)\right]$, $p={\left[p\left(x_{1}\right),\ldots ,p\left(x_{M}\right)\right]}^{T}={\left[\frac{e^{f(x_{1})}}{1+e^{f(x_{1})}},\ldots ,\frac{e^{f(x_{M})}}{1+e^{f(x_{M})}}\right]}^{T}$, and $y={\left[y_{1},\ldots ,y_{M}\right]}^{T}$. In addition, the Hessian operator is expressed as,
(15)$$\begin{array}{cc} H & =\frac{\partial g}{\partial f}=\frac{\partial }{\partial f}\left[\frac{e^{f(x_{1})}}{1+e^{f(x_{1})}},\ldots ,\frac{e^{f(x_{M})}}{1+e^{f(x_{M})}}\right]\Phi^{T}+\lambda I\\ & =\left[\frac{\partial }{\partial f}\frac{e^{f(x_{1})}}{1+e^{f(x_{1})}},\ldots ,\frac{\partial }{\partial f}\frac{e^{f(x_{M})}}{1+e^{f(x_{M})}}\right]\Phi^{T}+\lambda I\\ & =\left[\frac{e^{f(x_{1})}}{1+e^{f(x_{1})}}\frac{1}{1+e^{f(x_{1})}}\varphi \left(x_{1}\right),\ldots ,\frac{e^{f(x_{M})}}{1+e^{f(x_{M})}}\frac{1}{1+e^{f(x_{M})}}\varphi \left(x_{M}\right)\right]\Phi^{T}+\lambda I\\ & =\Phi \Lambda \Phi^{T}+\lambda I \end{array}$$
where I is an identity operator (or matrix) defined in the domain of H×H, so that for any function h∈H, and Ih=h. Here, Λ is an M×M diagonal matrix defined as,
(16)$$\Lambda =p{\left(1_{M}-p\right)}^{T}$$
and $1_{M}={\left[1,\ldots 1\right]}^{T}$ is an M×1 vector with all elements equal to one.

Using the Newton-Raphson method, the Hilbert function, is updated iteratively,
(17)$$\begin{array}{cc} f_{i+1} & =f_{i}-H_{i}^{-1}g_{i}=f_{i}-{\left[\Phi \Lambda_{i}\Phi^{T}+\lambda I\right]}^{-1}g_{i}\\ & =f_{i}-{\left[\Phi \Lambda_{i}\Phi^{T}+\lambda I\right]}^{-1}\left[\Phi \left(p_{i}-y\right)+\lambda f_{i}\right]\\ & =f_{i}-{\left[\Phi \Lambda_{i}\Phi^{T}+\lambda I\right]}^{-1}\Phi \left(p_{i}-y\right)-\lambda {\left[\Phi \Lambda_{i}\Phi^{T}+\lambda I\right]}^{-1}f_{i}\\ & \approx f_{i}-{\left[\Phi \Lambda_{i}\Phi^{T}+\lambda I\right]}^{-1}\Phi \left(p_{i}-y\right)\\ & =f_{i}-\Phi R_{i}\left(p_{i}-y\right) \end{array}$$
where $R_{i}={\left[\Lambda_{i}\Phi^{T}\Phi +\lambda I_{M}\right]}^{-1}$, $I_{M}$ is an M×M identity matrix, and the subscript “*i*” indicates the *i*th iteration for learning function *f*. The Hilbert function, f(x), is evaluated iteratively as follows,
(18)$$\begin{array}{cc} f_{i+1}\left(x\right) & ={\langle f_{i+1},\varphi \left(x\right)\rangle }_{H}\\ & =f_{i}\left(x\right)-\varphi^{T}\left(x\right)\Phi R_{i}\left(p_{i}-y\right)\\ & =f_{i}\left(x\right)-k^{T}R_{i}\left(p_{i}-y\right) \end{array}$$
where $k=\Phi^{T}\varphi \left(x\right)={\left[k\left(x_{1},x\right),\ldots ,k\left(x_{M},x\right)\right]}^{T}$.

It can be seen from (18) that the evaluation of $f_{i+1}\left(x\right)$ only depends on the evaluations of $f_{i}\left(x\right)$ and $k(x_{m},x)$, m=1,…,M. Therefore, the evaluation $f_{i}\left(x\right)$ for each iteration is not needed if $x\notin \{x_{1},\ldots ,x_{M}\}$. Instead, $f_{i}\left(x\right)$ can be calculated at the last iteration directly from the evaluations of $f_{i}\left(x_{m}\right)$, such that
(19)$$f_{i}\left(x\right)=f_{0}\left(x\right)-k^{T}\sum_{i^{\prime }=1}^{i}R_{i^{\prime }}\left(p_{i^{\prime }}-y\right)=f_{0}\left(x\right)-k^{T}S_{i}$$
where $f_{0}\left(x\right)$ is the initial function evaluation at x, prior to learning the function from M. The following matrix
(20)$$S_{i}≜\sum_{i^{\prime }=1}^{i}R_{i^{\prime }}\left(p_{i^{\prime }}-y\right)$$
only depends on the measurement data, M, and can be updated iteratively.

Furthermore, consider *Q* collocation points, $X^{c}={\left\{x_{q}^{c}\in W\right\}}_{q=1}^{Q}$, in the ROI, characterized by the same spatial interval, labeled by the superscript “c”. Let $\Phi^{c}=\left[\varphi \left(x_{1}^{c}\right),\ldots ,\varphi \left(x_{Q}^{c}\right)\right]$ denote the feature matrix of the collocation points. The evaluations of function $f(x_{q}^{c})$ at all collocation points can be updated by,
(21)$$f_{i+1}=f_{i}-{\Phi^{c}}^{T}\Phi R_{i}\left(p_{i}-y\right)$$
where $f_{i}={\Phi^{c}}^{T}f_{i}={\left[f_{i}\left(x_{1}^{c}\right),\ldots ,f_{i}\left(x_{Q}^{c}\right)\right]}^{T}$. According to (19), the evaluations, $f_{i}={\Phi^{c}}^{T}f_{i}$, can be updated directly by,
(22)$$f_{i}=f_{0}-{\Phi^{c}}^{T}\Phi S_{i}$$
where $f_{0}$ comprises the initial function evaluations at the collocation points prior to learning the function from the measurement data M.

In summary, instead of learning the function *f* or its coefficients as in traditional KLR or GPR [26], we update the evaluations of f(x) at the collocation points, $X^{c}$. Given the Hilbert function *f*, the evaluations at the collocation points provide the Hilbert map, f, defined as

(23)$$f≜{\Phi^{c}}^{T}f={\left[f\left(x_{1}^{c}\right),\ldots ,f\left(x_{Q}^{c}\right)\right]}^{T}$$

### 3.2. Temporal Update of Hilbert Map

The previous subsection develops a method for learning the Hilbert map from the measurement data set M, obtained by the sensors during one time interval. This subsection considers a Hilbert map updated according to the next data set, $M_{k}={\left\{\left(x_{k,m},y_{k,m}\right)\right\}}_{m=1}^{M_{k}}$, obtained during the *k*th time interval. Here, $M_{k}$ is the number of measurements in the *k*th time interval, and $X_{k}={\left\{x_{k,m}\right\}}_{m=1}^{M_{k}}$ and $Y_{k}={\left\{y_{k,m}\right\}}_{m=1}^{M_{k}}$ are the measurement positions and the corresponding classification estimates, respectively. To learn the next Hilbert map, the loss function over a period *T* can be expressed as,
(24)$$J_{T}=\sum_{k=1}^{T}\left[\gamma \left(T-k\right)\sum_{m=1}^{M_{k}}\ell_{k,m}\right]+\frac{\lambda }{2}{∥f∥}_{H}^{2}$$
where λ is the regularization term as in (13), γ(T-k) is the “forgetting factor”, and $\ell_{k,m}=-\ln\left[P(Y=y_{k,m}|x_{k,m})\right]$ is the NLL of the data $\left(x_{k,m},y_{k,m}\right)$. Similarly to (14), the gradient is expressed as,
(25)$$\begin{array}{cc} g_{T} & =\frac{\partial J_{T}}{\partial f}=\sum_{k=1}^{T}\left[\gamma \left(T-k\right)\sum_{m=1}^{M_{k}}\frac{\partial }{\partial f}\ell_{k,m}\right]+\lambda f\\ & =\left[\Phi_{1},\ldots ,\Phi_{T}\right]\Gamma_{T}{\left[{\left(p_{1}-y_{1}\right)}^{T},\ldots ,{\left(p_{T}-y_{T}\right)}^{T}\right]}^{T}\\ & ={\tilde{\Phi }}_{T}\Gamma_{T}{\left[{\left(p_{1}-y_{1}\right)}^{T},\ldots ,{\left(p_{T}-y_{T}\right)}^{T}\right]}^{T} \end{array}$$
where $\Phi_{k}=\left[\varphi \left(x_{k,1}\right),\ldots ,\varphi \left(x_{k,M_{k}}\right)\right]$, $p_{k}={\left[p\left(x_{k,1}\right),\ldots ,p\left(x_{k,M_{k}}\right)\right]}^{T}={\left[\frac{e^{f(x_{k,1})}}{1+e^{f(x_{k,1})}},\ldots ,\frac{e^{f(x_{k,M_{k}})}}{1+e^{f(x_{k,M_{k}})}}\right]}^{T}$, $y_{k}={\left[y_{k,1},\ldots ,y_{k,M_{k}}\right]}^{T}$ for k=1,…,T, and ${\tilde{\Phi }}_{T}=\left[\Phi_{1},\ldots ,\Phi_{T}\right]$. Furthermore, $\Gamma_{T}$ is a diagonal matrix obtained by placing the vector, ${\left[\begin{array}{ccccc} \gamma \left(T-1\right)1_{M_{1}}^{T} & \ldots & \gamma \left(T-k\right)1_{M_{k}}^{T} & \ldots & \gamma \left(0\right)1_{M_{T}}^{T} \end{array}\right]}^{T}$ on the diagonal of a zero matrix. In addition, as with (15), the Hessian operator can be expressed as,
(26)$$\begin{array}{cc} H_{T} & =\frac{\partial g_{T}}{\partial f}=\left[\Phi_{1},\ldots ,\Phi_{T}\right]\Gamma_{T}{\tilde{\Lambda }}_{T}{\left[\Phi_{1}^{T},\ldots ,\Phi_{T}^{T}\right]}^{T}+\lambda I\\ & ={\tilde{\Phi }}_{T}\Gamma_{T}{\tilde{\Lambda }}_{T}{\tilde{\Phi }}_{T}^{T}+\lambda I \end{array}$$
where,
(27)$${\tilde{\Lambda }}_{T}=\left[\begin{array}{ccccc} \Lambda_{1} & \ldots & 0 & \ldots & 0\\ \vdots & \ddots & & & \vdots \\ 0 & & \Lambda_{k} & & 0\\ \vdots & & & \ddots & \vdots \\ 0 & \ldots & 0 & \ldots & \Lambda_{T} \end{array}\right]$$
and $\Lambda_{k}=p_{k}{\left(1_{M_{k}}-p_{k}\right)}^{T}$.

Then, the update rule for learning the Hilbert function, *f*, at the *i*th iteration is given by,
(28)$$\begin{array}{cc} f_{T,i+1} & =f_{T,i}-H_{T,i}^{-1}g_{T,i}\\ & \approx f_{T,i}-{\left[{\tilde{\Phi }}_{T}\Gamma_{T}{\tilde{\Lambda }}_{T,i}{\tilde{\Phi }}_{T}^{T}+\lambda I\right]}^{-1}{\tilde{\Phi }}_{T}\Gamma_{T}{\left[{\left(p_{1,i}-y_{1}\right)}^{T},\ldots ,{\left(p_{T,i}-y_{T}\right)}^{T}\right]}^{T}\\ & =f_{T,i}-{\tilde{\Phi }}_{T}{\tilde{R}}_{i}\Gamma_{T}{\left[{\left(p_{1,i}-y_{1}\right)}^{T},\ldots ,{\left(p_{T,i}-y_{T}\right)}^{T}\right]}^{T} \end{array}$$
where ${\tilde{R}}_{i}={\left[\Gamma_{T}{\tilde{\Lambda }}_{T,i}{\tilde{\Phi }}_{T}\tilde{\Phi }+\lambda I_{M_{tol}}\right]}^{-1}$ and $M_{tol}=\sum_{k=1}^{T}M_{k}$. According to the above equation, the function, *f*, is expressed by using the set of all the measurement positions, $\left\{{⋃}_{k=1}^{T}X_{k}\right\}$, and the corresponding coefficients.

To reduce the computational load, the forgetting factor, γ(T-k), is modeled by,
(29)$$\gamma \left(T-k\right)=\left\{\begin{array}{cc} 1 & \mathrm{if}\;T-k<\tau \\ 0 & \mathrm{otherwise} \end{array}\right.$$
such that each robot stores only the measurements obtained during the past τ time steps. By setting τ=1, the update rule (28) for learning the function *f* at the *i*th iteration is expressed as,
(30)$$\begin{array}{cc} f_{T,i+1} & =f_{T,i}-{\left[\Phi_{T}\Lambda_{T}\Phi_{T}^{T}+\lambda I\right]}^{-1}\Phi_{T}\left(p_{T}-y_{T}\right)\\ & =f_{T,i}-\Phi_{T}R_{T,i}\left(p_{T,i}-y_{T}\right) \end{array}$$
where $R_{T,i}={\left[\Lambda_{i}\Phi_{T}^{T}\Phi_{T}+\lambda I_{M_{T}}\right]}^{-1}$.

### 3.3. Approximation of GDM

The previous subsections present a method for learning the Hilbert function, *f*, from the available measurement data, comprising the measurement position data set $X_{k}={\left\{x_{k,m}\right\}}_{m=1}^{M_{k}}$ and corresponding classification estimates $Y_{k}={\left\{y_{k,m}\right\}}_{m=1}^{M_{k}}$ for k=1,…,T that indicate whether the concentration measurement at the measurement position belongs to the range, R, defined in (9). If *L* classification estimates are obtained from the same concentration measurements, indicating that the concentration measurement belongs to the *L* concentration ranges, such as $R_{1},\ldots ,R_{L}$, respectively, using the learning method described in Section 3.2, one can approximate *L* Hilbert functions, $f_{1},\ldots ,f_{L}$. Furthermore, the probabilities in (6) can be evaluated at all the collocation points, $X^{c}={\left\{x_{q}\right\}}_{q=1}^{Q}$, such that

(31)$$p_{l}\left(x_{q}^{c}\right)=1-\frac{1}{e^{f_{l}\left(x_{q}^{c}\right)}},\;q=1,\ldots ,Q\;\mathrm{and}\;l=1,\ldots ,L$$

Now, consider Hilbert functions, ${\left\{f_{l}\right\}}_{l=1}^{L}$, approximated locally by each robot. Although all the concentration intervals are mutually exclusive and complete, i.e.,
(32)$$\begin{array}{cc} R_{i}\cap R_{j} & =\emptyset \end{array}$$
(33)$$\begin{array}{cc} \overset{L}{\underset{i=1}{\cup }}R_{i} & =[L_{c},H_{c}],\;i,j=1,\ldots ,L\;\mathrm{and}\;i\neq j \end{array}$$
it is not guaranteed that the sum of all the learned probabilities, ${\left\{p_{l}\left(x_{q}^{c}\right)\right\}}_{l=1}^{L}$, is equal to one, or $\sum_{l=1}^{L}p_{l}\left(x_{q}^{c}\right)=1$. Therefore, the learned probabilities must be normalized to obtain the discrete distribution, $\pi \left(x_{q}^{c}\right)=\left[\pi_{1}\left(x_{q}^{c}\right),\ldots ,\pi_{L}\left(x_{q}^{c}\right)\right]$. Each component of the discrete distribution is calculated by,
(34)$$\pi_{l}\left(x_{q}^{c}\right)=\frac{p_{l}\left(x_{q}^{c}\right)\prod_{1\leq l^{\prime }\neq l\leq L}\left[1-p_{l^{\prime }}(x_{q}^{c}\right]}{p_{0}\left(x_{q}^{c}\right)}\propto \frac{p_{l}\left(x_{q}^{c}\right)}{1-p_{l}\left(x_{q}^{c}\right)}$$
where $p_{0}\left(x_{q}^{c}\right)$ is a normalization term. Let $\bar{c}=\left[{\bar{c}}_{1},\ldots ,{\bar{c}}_{L}\right]$ denote the medians of these intervals, where ${\bar{c}}_{l}=\left(H_{l}-L_{l}\right)/2$. Then, the expectation of the random variable $C(x_{q}^{c})$ can be expressed as,
(35)$$E\left[C\left(x_{q}^{c}\right)\right]=\pi {\left(x_{q}^{c}\right)}^{T}\bar{c}$$
where E(·) is the expectation operator.

## 4. Information Fusion

In this section, the problem of information fusion between neighboring robots is considered, where each robot builds its own Hilbert function locally using a decentralized approach. Assume that all the robots use the same collocation points, $X^{c}$. To limit the amount of communicated data, the evaluations of the Hilbert functions at the collocation points are shared among the robots, instead of the coefficients and parameters of the Hilbert functions.

### 4.1. Hilbert Map Fusion

Consider two robots that have each learned their respective Hilbert functions, $f_{1}$ and $f_{2}$, from two different sets of measurement data, $M_{1}$ and $M_{2}$, respectively. Then, the fused Hilbert function, $f_{F}$, can be obtained based on the following theorem.

**Theorem** **1.**
*Let $f_{1}\left(x\right)$ and $f_{2}\left(x\right)$ be two Hilbert functions defined on a workspace W, and approximated by two robots based on their own measurement data sets, $M_{1}$ and $M_{2}$, respectively. These two Hilbert functions can be applied to calculate the conditional probability that the concentration is in the range R given the corresponding measurement data sets, as follows:*
(36)$$\begin{array}{cc} p_{1}\left(x|M_{1}\right) & =1-\frac{1}{1+e^{f_{1}\left(x\right)}} \end{array}$$
(37)$$\begin{array}{cc} p_{2}\left(x|M_{2}\right) & =1-\frac{1}{1+e^{f_{2}\left(x\right)}} \end{array}$$

*Then, the fused conditional probability $p_{F}\left(x|M_{1},M_{2}\right)$ can be expressed as*
(38)$$p_{F}\left(x|M_{1},M_{2}\right)=1-\frac{1}{1+e^{f_{F}\left(x\right)}}$$
*where $f_{F}\left(x\right)$ is the fused Hilbert function. In addition, the fused Hilbert function, $f_{F}\left(x\right)$, can be calculated from the Hilbert functions, $f_{1}\left(x\right)$ and $f_{2}\left(x\right)$, as follows,*
(39)$$f_{F}\left(x\right)=f_{1}\left(x\right)+f_{2}\left(x\right)-\ln\varepsilon$$
*where $\varepsilon =\frac{p(x)}{1-p(x)}$ is the ratio between the prior probabilities, P(C(x)∈R)=p(x) and P(C(x)∉R)=1-p(x), at x∈W.*


The proof of Theorem 1 is provided in the Section A.1. According to Theorem 1, the following corollary can be obtained.

**Corollary** **1.**
*Assume that the prior probability is even, such as p(x)=1/2. Then, the ratio between the prior probabilities can be calculated by,*
(40)$$\varepsilon =\frac{p(x)}{1-p(x)}=1$$
*and the fusion function can be rewritten as*
(41)$$f_{F}\left(x\right)=f_{1}\left(x\right)+f_{2}\left(x\right)$$


Because the prior concentration distribution is unknown, the Hilbert functions are fused according to (41) in Corollary 1, unless otherwise stated. Furthermore, assume that all the robots have the same collocation points, $X^{c}$. The information fusion can be implemented by fusing the Hilbert maps among neighboring robots,
(42)$$f_{H,F}=f_{H,1}+f_{H,2}$$
where $f_{H,1}$ and $f_{H,2}$, and $f_{H,F}$ are the Hilbert maps associated with the Hilbert functions $f_{1}\left(x\right)$, $f_{2}\left(x\right)$, and $f_{F}\left(x\right)$, respectively.

### 4.2. Communication Strategy

Any pair of robots in the network can share and update their Hilbert maps efficiently according to (42). To implement data fusion for a very large number of robots, however, a communication strategy is also required to determine how to share data between robots characterized by active communication links. In this paper, gas measurement data are communicated at every time step $T_{c}$. The communication protocol requires four steps at every time *k*, as follows. In the first, the *N* robots are partitioned into smaller communication networks according to their positions and communication range $r_{c}$, and $G_{ı}$ denotes the index set of the robot in the *ı*th communication network. Then, for any robot, $a_{n}$, with $n\in G_{ı}$, there exists another robot, $a_{n^{\prime }}$, such that
(43)$$d_{n,n^{\prime }}=∥x_{n}-x_{n^{\prime }}∥\leq r_{c},\;n,n^{\prime }\in G_{ı}$$
where $d_{n,n^{\prime }}$ is the distance between robots $a_{n}$ and $a_{n^{\prime }}$, and ∥·∥ indicates the Euclidean norm.

As a second step, one robot in every communication network is selected as a temporary data center (TDC) denoted by $a_{n_{ı}^{*}}$, $n_{ı}^{*}\in G_{ı}$. The other robots in $G_{ı}$ send the Hilbert map change $\Delta f_{n,k}$ to robot $a_{n_{ı}^{*}}$. The Hilbert map change, $\Delta f_{n,k}$, is defined as the change between the *n*th robot’s Hilbert map at the current time step *k* and its Hilbert map at the previous communication time step, $k-T_{c}$, such that
(44)$$\Delta f_{n,k}=f_{n,k}-f_{n,k-T_{c}},\;n\in G_{ı}$$
where $f_{n,k}$ and $f_{n,k-T_{c}}$ denote the Hilbert maps of the *n*th robot at times *k*th and $\left(k-T_{c}\right)$th, respectively.

In the third step, the sum of the Hilbert map changes obtained from all robots in $G_{ı}$, defined as,
(45)$$\Delta f_{G_{ı},k}≜\sum_{n\in G_{ı}}\Delta f_{n,k}$$
is communicated back to the other robots, except the TDC robot, $a_{n_{ı}^{*}}$. Finally, in the fourth step, all the robots update their own Hilbert maps by adding the received total Hilbert map changes, $\Delta f_{G_{ı},k}$, to the current Hilbert maps, such that
(46)$$f_{n,k^{+}}=f_{n,k}+\Delta f_{G_{ı},k},\;n\in G_{ı}$$
where $f_{n,k^{+}}$ represents the Hilbert map after the data fusion process.

## 5. Path-Planning Algorithms

In the previous sections, the Hilbert maps $f_{l,n,k}$, l=1,..,L, n=1,…,N and $k=1,\ldots ,T_{f}$, are approximated and updated by the robots. The corresponding binary probabilities $p_{l,n,k}$ can be calculated efficiently as follows
(47)$$\begin{array}{cc} p_{l,n,k} & ={\left[p_{l,n,k}\left(x_{1}^{c}\right),\ldots ,p_{l,n,k}\left(x_{Q}^{c}\right)\right]}^{T}\\ & ={\left[\frac{e^{f_{l,n,k}\left(x_{1}^{c}\right)}}{1+e^{f_{l,n,k}\left(x_{1}^{c}\right)}},\ldots ,\frac{e^{f_{l,n,k}\left(x_{1}^{c}\right)}}{1+e^{f_{l,n,k}\left(x_{1}^{c}\right)}}\right]}^{T} \end{array}$$
and the entropy at each collocation point $x_{q}^{c}\in X^{c}$, q=1,…,Q, is obtained by
(48)$$h_{l,n,k}\left(x_{q}^{c}\right)=-p_{l,n,k}\left(x_{q}^{c}\right)\log\left[p_{l,n,k}\left(x_{q}^{c}\right)\right]-\left[1-p_{l,n,k}\left(x_{q}^{c}\right)\right]\log\left[1-p_{l,n,k}\left(x_{q}^{c}\right)\right]$$

Then, an entropy map, $h_{n,k}$ is obtained from the vector,
(49)$$h_{n,k}={\left[h_{n,k}\left(x_{1}^{c}\right),\ldots ,h_{n,k}\left(x_{Q}^{c}\right)\right]}^{T}$$
where $h_{n,k}\left(x_{q}^{c}\right)=\sum_{l=1}^{L}h_{l,n,k}\left(x_{q}^{c}\right)$ denotes the sum of the entropy at the collocation point $x_{q}^{c}$.

According to the cost function in (5), the objective of the *n*th robot is to minimize the uncertainty of the concentration distribution, which can be implemented by minimizing the entropy at all the collocation points, such that

(50)$${\tilde{J}}_{n}\left(k\right)=\sum_{q=1}^{Q}h_{n,k}\left(x_{q}^{c}\right)$$

Therefore, ${\tilde{J}}_{n}\left(T_{f}\right)$ can be treated as an approximation of the term $\int_{W}H\left[C_{n}\left(x|M_{n}^{\left(T_{f}\right)}\right)\right]dx$ in (5). In the rest of paper, ${\tilde{J}}_{n}\left(T_{f}\right)$ is applied as the approximation of the final cost function in (5) unless otherwise stated. In other words, the *n*th robot should visit the area around the collocation point $x_{q}^{c}$, which has the higher value of $h_{n,k}\left(x_{q}^{c}\right)$ at the *k*th time step. Based on this idea, two entropy-based path-planning algorithms are proposed in the following section to control the robots such that the value of the concentration measurements obtained by the robots in the ROI can be optimized over time.

### 5.1. Entropy-Based Artificial Potential Field

An information-driven approach is developed by planning the path of the robots such that they move towards collocation points with higher entropy. The collocation points are treated as goal positions characterized by attractive artificial potential fields defined as,
(51)$$U^{att}\left(x_{n},x_{q}^{c}\right)=-\frac{h_{n,k}\left(x_{q}^{c}\right)}{∥x_{n}-x_{q}^{c}{∥}^{2}},\;q=1,\ldots ,Q\;\mathrm{and}\;n=1,\ldots ,N$$
where the superscript “att” indicates the attractive field. The corresponding attractive gradient is expressed as
(52)$$g^{att}\left(x_{n},x_{q}^{c}\right)=\frac{\partial U_{q}^{att}\left(x_{n}\right)}{\partial x_{n}}=\frac{U_{q}^{att}\left(x_{n}\right)}{∥x_{n}-x_{q}^{c}{∥}^{4}}\left(x_{n}-x_{q}^{c}\right),\;q=1,\ldots ,Q$$

Similarly to classic artificial potential field methods, the repulsive potential functions $U^{rep}$ generated from the other robots are also considered, such that
(53)$$U^{rep}\left(x_{n},x_{n^{\prime }}\right)=\left\{\begin{array}{cc} \frac{1}{2}{\left(\frac{1}{∥x_{n}-x_{n^{\prime }}∥}-\frac{1}{r_{rep}}\right)}^{2}, & ∥x_{n}-x_{n^{\prime }}∥\leq r_{rep}\\ 0, & ∥x_{n}-x_{n^{\prime }}∥>r_{rep} \end{array}\right.,\;1\leq n\neq n^{\prime }\leq N$$
where $r_{rep}$ is the distance threshold to create a repulsion effect on the robot. The repulsive gradient is expressed as
(54)$$g^{rep}\left(x_{n},x_{n^{\prime }}\right)=\left\{\begin{array}{cc} -\left(\frac{1}{∥x_{n}-x_{n^{\prime }}∥}-\frac{1}{r_{rep}}\right)\frac{x_{n}-x_{n^{\prime }}}{∥x_{n}-x_{n^{\prime }}{∥}^{3}}, & ∥x_{n}-x_{n^{\prime }}∥\leq r_{rep}\\ 0, & ∥x_{n}-x_{n^{\prime }}∥>r_{rep} \end{array}\right.,\;1\leq n\neq n^{\prime }\leq N$$

Using (52) and (54), the total potential gradient for the *n*th robot is expressed as,
(55)$$g^{tol}\left(x_{n}\right)=\xi \sum_{q=1}^{Q}g^{att}\left(x_{n},x_{q}^{c}\right)+\eta \sum_{1\leq n^{\prime }\neq n\leq N}g^{rep}\left(x_{n},x_{n^{\prime }}\right)$$
where ξ and η are user-defined coefficients.

Based on the total gradient, $g^{tol}\left(x_{n}\right)$, the *n*th robot can be controlled to visit the collocation points with higher uncertainty and avoid collisions with other robots. The algorithm is developed based on the entropy attraction, which is named as EAPF algorithm.

### 5.2. Entropy-Based Particle Swarm Optimization

The particle swarm optimization (PSO) algorithm proposed by Clerc and Kennedy in [33] and its variants use a “constriction coefficient” to prevent the “explosion behavior” of the particles, and have been successfully applied to GDM and gas source localization problems [12,13]. In the original PSO algorithm and its variants, the concentration measurements are used to update the robot controls. Considering the different objective function in (5), an entropy-based PSO (EPSO) is proposed.

At the *k*th time step, the update of the *n*th robot position, $x_{n,k}$, can be described as
(56)$$\begin{array}{cc} \nu_{n,k} & =\chi \left[\nu_{n,k-1}+\psi_{1}g^{att}\left(x_{n,k},b_{n,k}^{neig}\right)+\psi_{2}g^{att}\left(x_{n,k},b_{n,k}^{glob}\right)\right]+\eta \sum_{1\leq n^{\prime }\neq n\leq N}g^{rep}\left(x_{n,k},x_{n^{\prime },k}\right)\\ x_{n,k+1} & =x_{n,k}+\nu_{n,k} \end{array}$$
where $\nu_{n,k}$ represents the velocity of the *n*th robot at the *k*th time step, $b_{n,k}^{neig}$ and $b_{n,k}^{glob}$ are the best neighboring and global collocation points, respectively. The best neighboring collocation point, $b_{n,k}^{neig}$, is determined by,
(57)$$b_{n,k}^{neig}=\underset{x\in X^{c},\;∥x_{n,k}-x∥\leq r_{neig}}{arg max}h_{n,k}\left(x\right)$$
where $r_{neig}$ is a coefficient which specifies the neighbor area from $x_{n,k}$. The best global collocation point, $b_{n,k}^{glob}$, is determined by
(58)$$b_{n,k}^{glob}=\underset{x\in X^{c}}{arg max}h_{n,k}\left(x\right)$$

The learning coefficients, $\psi_{1}\in \left[0,{\bar{\psi }}_{1}\right]$ and $\psi_{2}\in \left[0,{\bar{\psi }}_{2}\right]$, are two uniform random variables. The constant parameter χ>0 prevents the explosion behavior. For efficient performance and prevention of the explosion behavior in (56), the parameter settings of the learning coefficients proposed in [12,13,33] is applied. The constriction parameter χ>0 is calculated by (refer to [33]),
(59)$$\chi =\left\{\begin{array}{cc} \frac{2\kappa }{\psi -2+\sqrt{\psi^{2}-4\psi }}, & \;\mathrm{if}\;\psi >4\\ \kappa , & otherwise \end{array}\right.$$
where $\psi ={\bar{\psi }}_{1}+{\bar{\psi }}_{2}$ and κ∈[0,1].

## 6. Simulations and Results

The performance of the decentralized GDM methods presented in this paper is demonstrated on a gas sensing application, where information about gas concentration obtained by a large network of robots is fused and used for information-driven path planning in a decentralized approach. Two indoor and outdoor environments are simulated and used to test the proposed methods. The new entropy-based EAPF and EPSO path-planning algorithms are compared to the existing algorithms known as random walk (RW) and classical particle swarm optimization (CPSO) [12,13,33].

### 6.1. Indoor GDM Sensing

The decentralized sensing system consists of a network of N=100 robots characterized by single integrator dynamics,
(60)$${\dot{x}}_{i}=u_{i},\;i=1,\ldots ,N$$
where $x_{i}={\left[x,\;y\right]}^{T}$ is the robot state vector, x,y are the inertial coordinates, and $u_{i}={\left[u_{1},\;u_{2}\right]}^{T}$ is the robot control vector comprised of the *x*- and *y*-velocity components. The robot state/position is assumed to be observable by a built-in GPS with zero-mean measure noise *w*, where *w* is Gaussian white noise N(0,Σ) with $\Sigma =0.05\times I_{2}$. The above distributed sensing system is tasked with mapping a gas distribution in an ROI $W=\left[0,L_{x}\right]\times \left[0,L_{y}\right]$ where $L_{x}=200$ m and $L_{y}=160$ m, with an unknown gas distribution shown in Figure 1, and over a fixed time interval $\left[0,\;T_{f}\right]$, where $T_{f}=500$ min. The normalized gas concentration range R=[0,100] (Figure 1) is divided into L=3 intervals: $R_{1}=\left[0,30\right)$, $R_{2}=\left[30,70\right)$, and $R_{3}=\left[70,100\right]$, representing low-hazard, medium-hazard, and high-hazard concentrations, respectively.

The robots are initially deployed at four corners in the ROI by sampling from a given Gaussian Mixed Model with 4 components where $\mu_{1}={\left[20,20\right]}^{T}$, $\mu_{2}={\left[20,140\right]}^{T}$, $\mu_{3}={\left[180,20\right]}^{T}$ and $\mu_{4}={\left[180,140\right]}^{T}$, and identical covariance matrices, $\Sigma =\left[\begin{array}{cc} 10 & 0\\ 0 & 10 \end{array}\right]$. The initial robot positions are shown as red points in Figure 1. Each robot is equipped with a small metal oxide sensor array comprised by M=5×5=25 gas sensors, where the spatial intervals between two sensors are all 10 cm. The FOV of each sensor array covers an area of 40×40 cm${}^{2}$ in the ROI, which is very small relative to the whole ROI. Assume that the measurement noise of the gas concentration V(x) is also characterized by Gaussian white noise, $N(0,\Sigma_{c}\left(x\right))$ with the covariance matrix $\Sigma_{c}\left(x\right)=0.5\times I_{2}$ everywhere in W. All of robots communicate with each other at the same time with a communication period $T_{c}=10$ min. The communication range is $r_{c}=30$ m. In addition, Q=100×100 virtual collocation points are evenly deployed in the ROI to generate the Hilbert maps, where the Gaussian kernel is used for Hilbert function learning with a kernel size of σ=2 m, and the parameter τ is set to 3.

The EPSO and the CPSO neighbor range coefficient, $r_{neig}=5$ m, is applied to determine the best neighboring collocation points. The maximum velocity of each robot is 2 m/min for all simulations. The four path-planning algorithms, referred to as EAPF, EPSO, CPSO, and RW, are tested for mapping the gas distribution. The approximated cost function ${\tilde{J}}_{n}\left(k\right)$ defined in (50) is calculated by each robot at every time step as shown in Figure 2, where the solid line represents the mean of the approximated cost values over all of the robots at the *k*th time step, and the dashed line indicates one standard deviation above and below the mean. These cost histories reflect how effective the robots are at mapping the gas distribution. A lower value of ${\tilde{J}}_{n}\left(k\right)$ indicates that more information about the gas distribution is obtained by the robot at the *k*th step. It can be observed that the EAPF and EPSO algorithms achieve significantly better performance than the others. They can map the gas distribution more rapidly and completely, while the CPSO and RW algorithms perform poorly because they do not used prior measurement data for planning.

The means and standard deviations of the approximated cost function values of the all robots at the final time $k=T_{f}=500$ for all the algorithms are tabulated in Table 1. It can be seen that the approximated cost function obtained by the EPSO algorithm outperforms the other algorithms in the indoor environment. Let $n^{*}$ denote the index of the robot who gets the lowest value of ${\tilde{J}}_{n}\left(T_{f}\right)$ at the final step. Then, the value of ${\tilde{J}}_{n^{*}}\left(k\right)$ represents the best performance among all the robots. The estimates of the gas concentration in the ROI are calculated based on these learned Hilbert maps according to (35), where $\bar{c}=\left[15,50,85\right]$ is calculated from the cutoff coefficients. For each path-planning algorithm, three snapshots of the estimated GDMs generated by the $n^{*}$th robot in each simulation are presented in Figure 3, where the snapshots are taken at the k=20th, k=100th, k=500th time steps, respectively. It can be seen that the robots controlled by the EAPF and EPSO algorithms obtain clean GDMs at the final time step, while the robots controlled by the other two existing algorithms cannot complete the mapping task in the given time period.

The normalized mean square errors (NMSE) between the estimated gas distribution and the actual gas distribution are calculated for each robot. The means and the corresponding standard deviations of the NMSE over the all robots for the different planning algorithms are reported in Table 2, which obviously shows that the EPSO algorithm outperforms the other algorithms to estimate the GDMs.

### 6.2. Outdoor GDM

To verify the robustness and versatility of the proposed approaches, a GDM shown in Figure 4, originally presented in [34], is used to represent the gas distribution in an outdoor environment. The gas concentrations are normalized to the range R=[0,100] for comparison like the indoor simulations. In this case, the intervals are chosen as $R_{1}=\left[0,60\right)$, $R_{2}=\left[60,80\right)$, and $R_{3}=\left[80,100\right]$, to represent the plume shapes. All other parameters, including the initial robot positions, are the same as those used in the previous subsection.

The approximated cost function ${\tilde{J}}_{n}\left(k\right)$ obtained by all four algorithms is plotted in Figure 5. The approximated cost function at the final step are also tabulated in Table 3. Similarly to the indoor simulations, the gas concentration is estimated according to (35), where $\bar{c}=\left[30,70,90\right]$ is calculated based on the cutoff coefficients. The evolution of the estimated GDM obtained by different algorithms is presented in Figure 6. Furthermore, the statistical results of the NMSE between the estimated GDM and the actual GDM are reported in Table 4.

As expected, the results presented in Figure 5 and Figure 6 and Table 3 and Table 4 all show that the proposed EAPF and EPSO algorithms work well in the outdoor GDM problem, while the CPSO and RW algorithms cannot map the gas concentration in the entire workspace in the given time period. In addition, the EPSO algorithm significantly outperforms the other algorithms in all simulations.

## 7. Conclusions

This paper presents a decentralized framework for GDM and information-driven path planning in distributed sensing systems. GDM is performed using a probabilistic representation known as a Hilbert map and a novel Hilbert map fusion method is presented that quickly and efficiently combines information from many neighboring robots. In addition, two entropy-based path-planning algorithms, namely the EAPF and EPSO algorithms, are proposed to efficiently control all the robots to obtain the gas concentration measurements in the ROI. The proposed approaches are demonstrated on a system with hundreds of robots that must map a gas distribution collaboratively over a large ROI using on-board iron-oxide arrays and no prior information. The results show that through fusion and decentralized processing, the entropy of the gas map decreases over time, the robot paths remain safe (avoiding mutual collisions), and the entropy-based methods far outperform both traditional and random approaches.

## Figures and Tables

**Figure 1 sensors-19-01524-f001:**
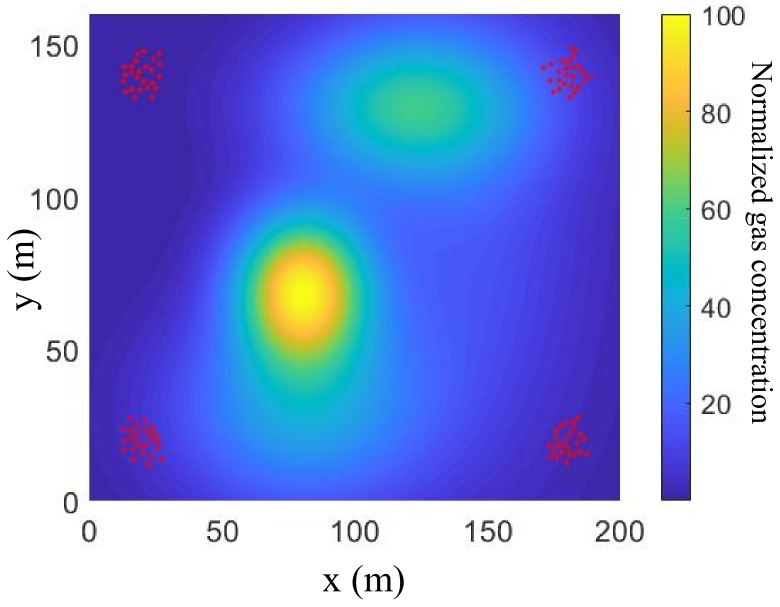
Gas concentration distribution in an indoor environment and initial robot deployment in ROI (plotted by red points).

**Figure 2 sensors-19-01524-f002:**
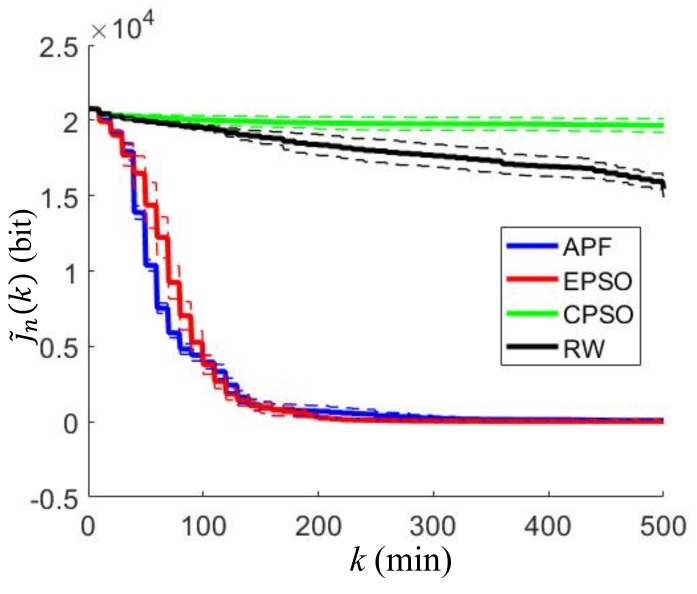
Approximated cost functions for different path-planning algorithms in the indoor environment, where the solid lines represent the mean of the approximated cost values over all the robots at the *k*th time step and the dashed lines indicate one standard deviation above and below the solid lines.

**Figure 3 sensors-19-01524-f003:**
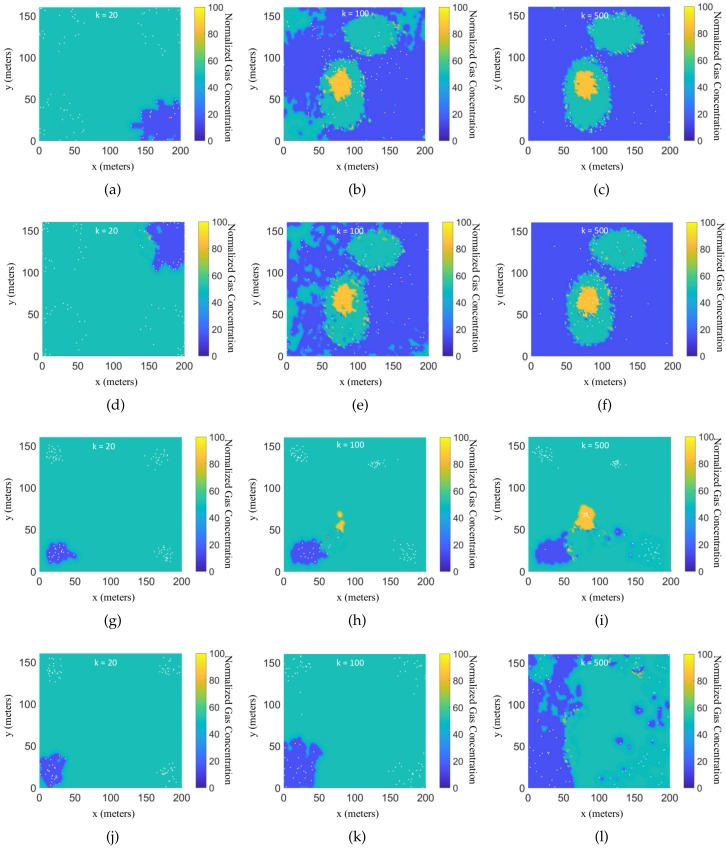
Evolution of the estimated gas distribution map in the indoor environment generated by the $n^{*}$th robot in each simulation at three instants in time, where the red point indicates the position of the $n^{*}$th robot and white points indicate the other robots, where the sub-figures in the first row (**a**–**c**) show the evolution of the estimated gas distribution obtained by the EAPF algorithm; the sub-figures in the second row (**d**–**f**) show the evolution of the estimated gas distribution obtained by the EPSO algorithm; the sub-figures in the third row (**g**–**i**) show the evolution of the estimated gas distribution obtained by the CPSO algorithm; and the sub-figures in the fourth row (**j–l**) show the evolution of the estimated gas distribution obtained by the RW algorithm.

**Figure 4 sensors-19-01524-f004:**
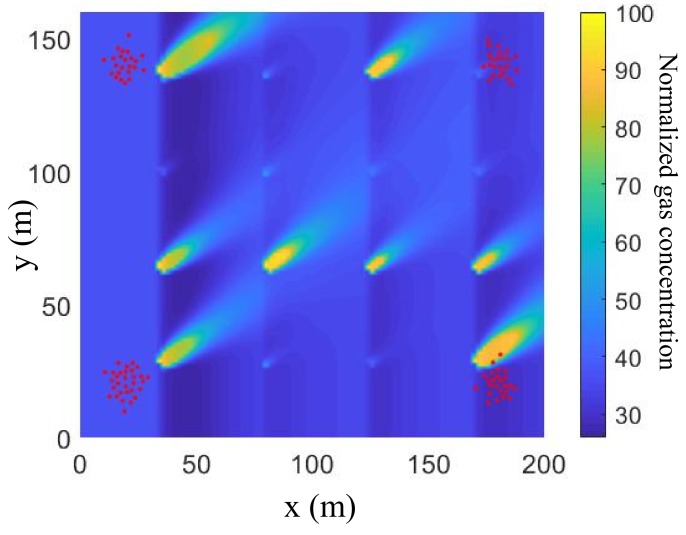
ROI and Gas concentration distribution in an outdoor environment and initial robot deployment in the ROI, where the red points represent the robots.

**Figure 5 sensors-19-01524-f005:**
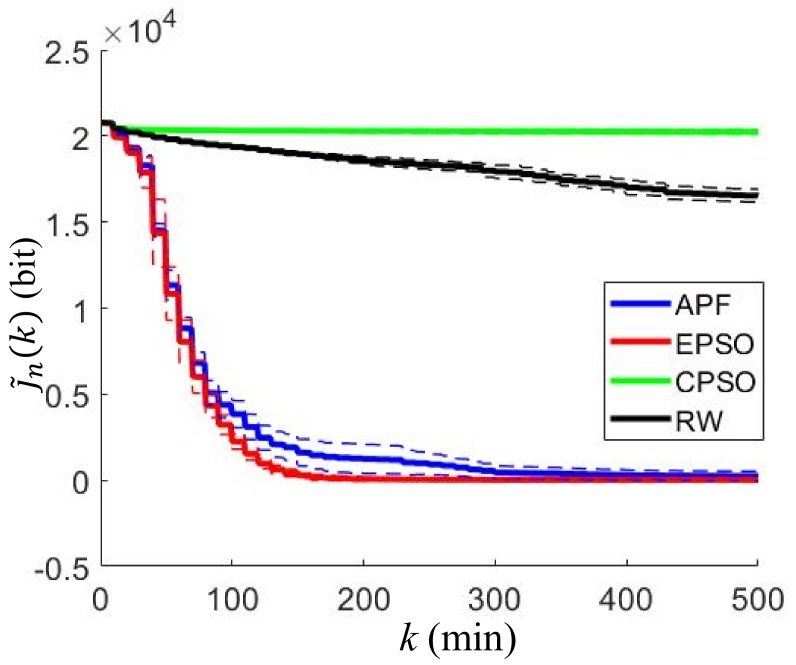
Approximated cost functions for different path-planning algorithms in the outdoor environment, where the solid lines represent the mean of the approximated cost values over all the robots at the *k*th time step and the dashed lines indicate one standard deviation above and below the solid lines.

**Figure 6 sensors-19-01524-f006:**
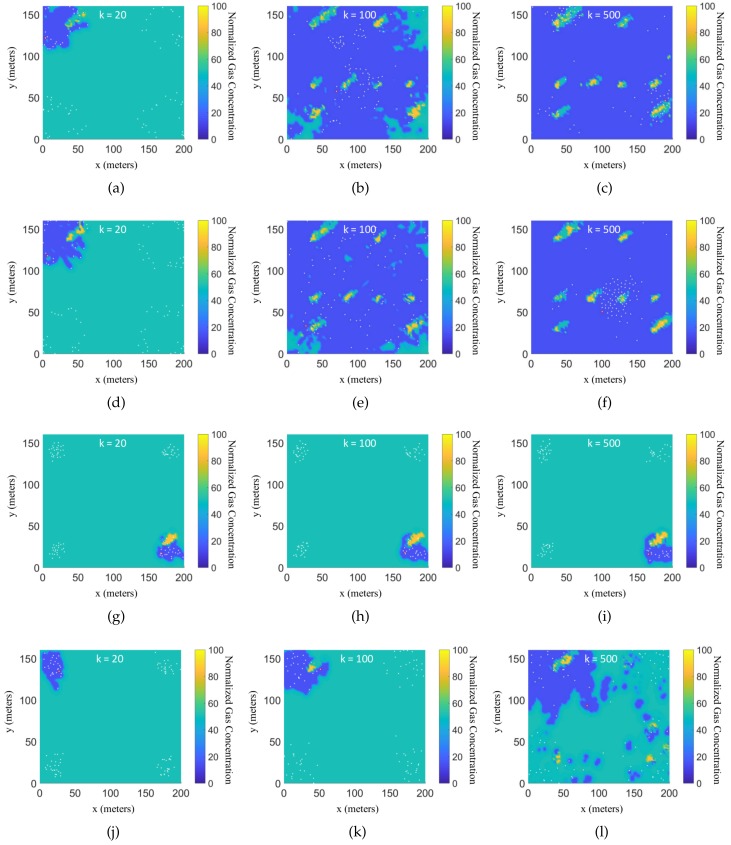
Evolution of the estimated gas distribution map in the outdoor environment generated by the $n^{*}$th robot in each simulation at three instants in time, where the red point indicates the position of the $n^{*}$th robot and white points indicate the other robots, where the sub-figures in the first row (**a**–**c**) show the evolution of the estimated gas distribution obtained by the EAPF algorithm; the sub-figures in the second row (**d**–**f**) show the evolution of the estimated gas distribution obtained by the EPSO algorithm; the sub-figures in the third row (**g**–**i**) show the evolution of the estimated gas distribution obtained by the CPSO algorithm; and the sub-figures in the fourth row (**j**–**l**) show the evolution of the estimated gas distribution obtained by the RW algorithm.

**Table 1 sensors-19-01524-t001:** Statistical results of the approximated cost function at the final time step in the indoor environment.

Algorithm	Mean of ${\tilde{J}}_{n}\left(T_{f}\right)$	Std. of ${\tilde{J}}_{n}\left(T_{f}\right)$
EAPF	72.01	66.64
EPSO	28.003	36.56
CPSO	19650.11	459.56
RW	15442.17	555.88

**Table 2 sensors-19-01524-t002:** Statistical results of the NMSE between the estimated GDMs and the actual GDMs at the final time step in the indoor environment.

Algorithm	Mean of NMSE	Std. of NMSE
EAPF	0.17521	0.00934
EPSO	0.17022	0.00432
CPSO	1.72230	0.03225
RW	1.20700	0.12516

**Table 3 sensors-19-01524-t003:** Statistic results of the approximated cost function at the final time step in the outdoor environment.

Algorithm	Mean of ${\tilde{J}}_{n}\left(T_{f}\right)$	Std. of ${\tilde{J}}_{n}\left(T_{f}\right)$
EAPF	243.9842	263.3285
EPSO	19.7781	55.7657
CPSO	20232.4784	147.2573
RW	16510.9489	380.798

**Table 4 sensors-19-01524-t004:** Statistic results of the estimated gas distribution maps at final time step in the outdoor environment.

Algorithm	Mean of NMSE	Std. of NMSE
EAPF	0.05368	0.00506
EPSO	0.04272	0.00123
CPSO	0.51741	0.00402
RW	0.41734	0.01094

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
