# Peer review of "Scalable Gas Sensing, Mapping, and Path Planning via Decentralized Hilbert Maps"

_sensors, 2019, doi:10.3390/s19071524_

Round 1

Reviewer 1 Report

The paper proposes a method for decentralised gas distribution mapping using Hilbert map fusion in very large scale robotics systems. The novelty from the algorithmic point of view is the fusion method for combining maps created by different robots. The novelty in the field of gas sensing with mobile robotics places in the fact that the authors propose a decentralised approach to this problem. The paper is well written and has a sound approach to solve the problem. The main comment is that the paper hasn't compared the proposed solution with any of the existing solutions from the perspective of the quality of the map created. Moreover, recent literature in the field of gas sensing in mobile robotics has not been cited. 

More detailed comments:

- The authors haven't explained their definition of "time-constant". I assume they meant situation were gas dispersion stays constant, which is a strong and unrealistic assumption in a real-world application. Especially because even we assume that external factors don't exist, e.g. in a closed room, gas dispersion varies by time due to turbulence. In that sense, the proposed approach fits better occupancy map scenarios rather than gas distribution mapping. What makes this solution appropriate in particular for gas distribution mapping?

- The paper lacks insight about gas sensors which is needed in developing solutions for gas distribution mapping. For example, each measurement of traditional gas sensors, e.g. metal oxide sensors, covers a few centimetres in space. Clearly, such sensors are not suitable for mapping a very large area. In case, the authors have LIDAR sensors in mind, e.g. TDLAS, the question would be how the authors want to deal with reflective surface needed.

- Usually, the presence of the gas is not zero and one, but it rather a range of probabilities. Isn't another simplifying assumption to formulate the problem as a binary classification task? Why authors didn't use a discretised relative percentage of the gas concentration? 

- Is the order of communication between robots effects the quality of the centralized map obtained? If the robots move randomly in space how fast do you expect to have a centralized map?

- All the results are only shown in one scenario which makes it hard to understand how much the results are generalizable. For example, why the performance of the system considering 100 and 150 agents are so similar, however, there is a quite different behaviour when the number of agents is 200. What is the reason behind this difference? It is not clear to me why the relationship between the percentage of robot communication and performance coefficient is linear? Could it be because of the way that robots move relative to each other? 

Reviewer 2 Report

The authors present a collaborative algorithm to map gas concentration in a region of interest. The topic is interesting for the community. The paper and results are well presented.

However,  the authors fall short in results. They present only simulations. The approach should be validated in  a real task, that includes many more difficulties than a simulated arena. If the work is purely numeric, authors, at least, should quantify the benefits of the proposed approach with respect to other approaches, in terms of speed, resources needed, accuracy, etc...

Moreover, the authors claim in the abstract that they do not need prior information. Actually, they assume there is only one gas in the mapping. What assumptions are needed? At least, they assume static plume, with no bouts (unrealistic scenario). What is the time constant of the sensors? Finally, authors claim obstacle avoidance in the abstract. I did not see any obstacle in the simulated arenas.

Round 2

Reviewer 1 Report

The authors have considerably updated the results and addressed the comments. At this point, I will recommend this paper for publication.

Reviewer 2 Report

the authors successfully implemented my comments. I recommend the publication of the manuscript.